

# Diatom identification including life cycle stages through morphological and texture descriptors

Carlos Sánchez[1], Gabriel Cristóbal[1] and Gloria Bueno[2]

[1] Instituto de Óptica "Daza de Valdés", CSIC, Madrid, Spain
[2] VISILAB, Universidad de Castilla La Mancha, Ciudad Real, Spain

## ABSTRACT

Diatoms are unicellular algae present almost wherever there is water. Diatom identification has many applications in different fields of study, such as ecology, forensic science, etc. In environmental studies, algae can be used as a natural water quality indicator. The diatom life cycle consists of the set of stages that pass through the successive generations of each species from the initial to the senescent cells. Life cycle modeling is a complex process since in general the distribution of the parameter vectors that represent the variations that occur in this process is non-linear and of high dimensionality. In this paper, we propose to characterize the diatom life cycle by the main features that change during the algae life cycle, mainly the contour shape and the texture. Elliptical Fourier Descriptors (EFD) are used to describe the diatom contour while phase congruency and Gabor filters describe the inner ornamentation of the algae. The proposed method has been tested with a small algae dataset (eight different classes and more than 50 samples per type) using supervised and non-supervised classification techniques obtaining accuracy results up to 99% and 98% respectively.

Corresponding author
Carlos Sánchez,
carlos.sanchez@io.cfmac.csic.es

## INTRODUCTION

Diatoms or *Bacillariophyceae* are a group of unicellular algae distributed in a great variety of aquatic environments around the world. It has been estimated that there are more than 200,000 different species (*Mann & Droop, 1996*), each of them adapted to certain autoecological ranges. Such number was ellucidated taking into account three factors: the number of species already described (around 10.000), the use of a coarse-grained taxonomic approach and the number of understudied habitats. Other authors provide a more conservative number of 20.000 species (*Guiry, 2012*). Therefore, a clear relationship can be established between the composition of the diatom community and the physicochemical parameters of the environment in which they are developed. In environmental studies, algae can be used as a natural water quality indicator. Since 2004, the European directive (*European Committee for Standardization, 2004*) has established algae indices as a measurement of water quality for rivers, lakes, etc. Due to the diatom silica nature, their fossils can also be used for palaeoenvironmental studies.

Diatoms are formed by a silica capsule also known as frustule. There are different frustule shapes, like rounded (centric diatoms) or elongated algae (pennate diatoms). Between the pennate diatoms there are two different classes depending on the presence or absence of the raphe. The frustule is a siliceous covering formed by two elements (thecae) that fit together enveloping the cell. In most pennate diatoms, each thecae is traversed longitudinally by a groove (raphe) divided into two branches by a central area, where occasionally one or more slits appear (stigmas). Perpendicular to the raphe numerous striae formed by the alignment of several pores are arranged. Diatoms have two different reproduction stages, asexual and sexual. On the one hand, in the asexual stage, the cell separates both valves and it grows the other half resulting in two different algae, one being bigger than the other. This size change is what is called a life cycle that is formed by all the diatom generations. On the other hand, when the algae reaches a critical size where it cannot be reduced, the sexual reproduction and auxospore formation takes part. The auxospores form a new full size algae that will start the process again.

Traditionally, diatom identification has been made by expert biologists. They usually use morphometric measures, such as length and width, and other frustule characteristics, like striae density, and they make the identification comparing specimens with previously described diatom in the literature (*Blanco, Borrego-Ramos & Olenici, 2017*). This task is challenging due to a huge number of diatom species, similarities between species and life cycle related changes in shape and texture. Other researchers (*Pappas & Stoermer, 2003*) used shape descriptors based on Legendre polynomials and principal component analysis (PCA) in the identification of the *Cymbella cistula* species. (*Mou & Stoermer, 1992*) Applied PCA to the Fourier descriptors extracted from the contour of the *Tabellaria* group. There are also recent studies on the application of different classification methodologies and the consideration of different image features such as textures (*Coste et al., 2009*), geometry, morphology (*Falasco et al., 2009a*; *Cejudo-Figueiras et al., 2011*; *Woodard & Neustupa, 2016*; *Woodard et al., 2016*), contour analysis (*Kloster, Kauer & Beszteri, 2014*), combination of the above mentioned features (*Bueno et al., 2017*) and convolutional neural networks (*Pedraza et al., 2017*).

In this paper, we present a different approach considering the main features that change during algae life cycle, mainly the contour shape but also the texture. The life cycle consists of the set of stages that pass through the successive generations of each diatom species from the initial to the senescent cells. Life cycle modeling is a complex process. In general, the distribution of the parameter vectors representing the variations that occur in this process is non-linear and of high dimensionality. *Hicks et al. (2002)* analyzed several methods of diatom life cycle modeling, selecting among them the main curves method. However, it remains a challenging topic still open to new contributions. To date, there is no system capable of model variations in both the contour and the texture of a relatively large number of species (*Hicks et al., 2006*). One key reason is due to the difficulty of capturing a sufficient number of specimens of each species in each of the stages of its life cycle. In this work, we model the algae contour using the Elliptical Fourier Descriptors (EFD) (*Kuhl & Giardina, 1982*). EFD have been widely used to describe closed curves as in (*Iwata & Ukai, 2002*) or more specifically to describe diatom contour (*Dimitrovski et al., 2012*). Invariance under

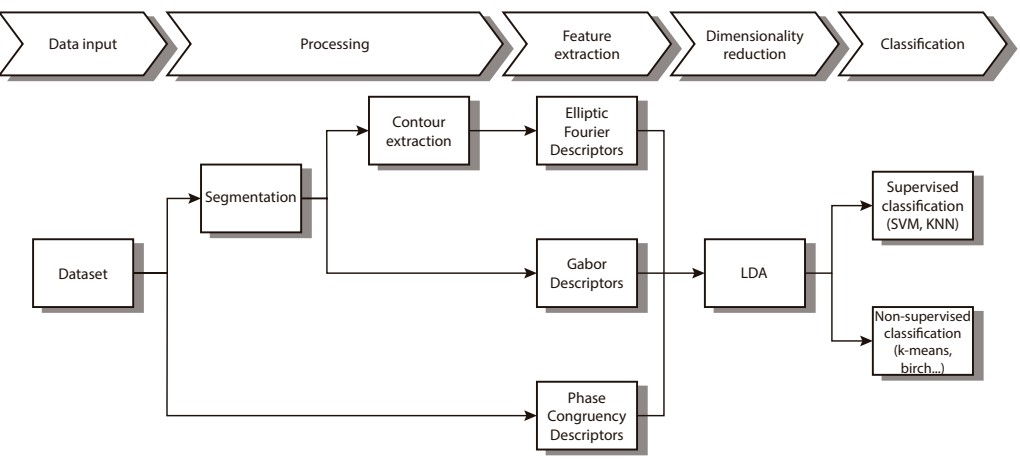

**Figure 1   Workflow of the proposed method.**

translation, scaling and rotation is achieved with EFD. To describe the texture of the valve we have chosen two different features that has been proven to work well in bright field microscopy images, that is, phase congruency and Gabor filter with statistical features. Phase congruency descriptors have been used to obtain robust edge detection (*Sosik & Olson, 2007*) and to identify phytoplankton. Gabor features have shown a high degree of discriminability in diatom classification (*Bueno et al., 2017*).

## MATERIALS AND METHODS

In this paper we propose a new combination of image features that allow us to automatically distinguish between different diatom. As Fig. 1 shows, we first segment and obtain the contour. After that, different features that characterize the contour and the texture are extracted followed by a reduction of dimensionality of the feature vector. The last step implements both supervised and non-supervised multivariate techniques to categorize the different taxa.

### Database

Sample images corresponding to the AQUALITAS project in Table 1 were captured using a low cost Brunel SP30 monocular microscope with standard Brunel DIN parfocal objectives of 60× (0.85 NA) and 100× (1.25 NA) using a LED with white light ($\lambda = 442$ nm). A Brunel Digicam LCMOS 5 Mpixel camera was used for image acquisition (cell size 2.2 $\mu$m × 2.2 $\mu$m). The camera is connected to the computer through an USB2.0 connection, providing an image size of 2,592 × 1,944 pixels. Images corresponding to (*Mann & Bayer, 2018*; *Blanco, Borrego-Ramos & Olenici, 2017*) in Tables 1– 3 have been provided by the authors.

Images from Table 1 form the dataset used in this paper for a total of 703 images of eight different classes. Figure 2 shows an example of the dataset. Additionally, images from two other datasets (Tables 2 and 3) have been tested. 382 images of the genera *Sellaphora* corresponding to six different taxa and 244 of the genera *Gomphonema* of five different taxa

**Table 1  Number of images in the dataset.**

| Taxa | #valves |
| --- | --- |
| *Gomphonema minutum*[a] | 74 |
| *Luticola goeppertiana*[a] | 117 |
| *Nitzschia amphibia*[a] | 59 |
| *Nitzschia capitellata*[a] | 95 |
| *Eunotia tenella*[b] | 68 |
| *Fragilariforma bicapitata*[b] | 100 |
| *Gomphonema augur var augur*[b] | 98 |
| *Stauroneis smithii grunow*[b] | 92 |

**Notes.**
[a] Images from AQUALITAS dataset, available at *Blanco (2018a)*.
[b] Images from DIADIST dataset *Mann & Bayer (2018)*.

**Table 2  Number of images per taxa in *Mann et al. (2004)* dataset.**

| Taxa | #valves |
| --- | --- |
| *Sellaphora pupula* | 40 |
| *Sellaphora obesa* | 72 |
| *Sellaphora blackfordensis* | 57 |
| *Sellaphora capitata* | 120 |
| *Sellaphora auldreekie* | 40 |
| *Sellaphora lanceolata* | 53 |

**Table 3  Number of images per taxa in *Blanco, Borrego-Ramos & Olenici (2017)* dataset, available at *Blanco (2018b)*.**

| Taxa | #valves |
| --- | --- |
| *Gomphonema acidoclinatum* | 76 |
| *Gomphonema auritum* | 40 |
| *Gomphonema gracile* | 28 |
| *Gomphonema jadwigiae* | 72 |
| *Gomphonema parvulum var parvulum* | 28 |

respectively. Dataset 1 is the only one of the three datasets that include more morphological variability in the species due to the presence of different stages in the diatom life cycle.

## Segmentation and contour extraction

It is needed to compute a binary mask of the diatom for extracting Gabor descriptors and the Fourier descriptors from the diatom contour.

The algorithm for mask extraction is as follows (available in *Sanchez, 2019*):

1. Binarization of the image using Otsu method to select the threshold by minimizing the intraclass variance between white and black pixels.
2. Image dilation.
3. Hole filling.
4. Image erosion.

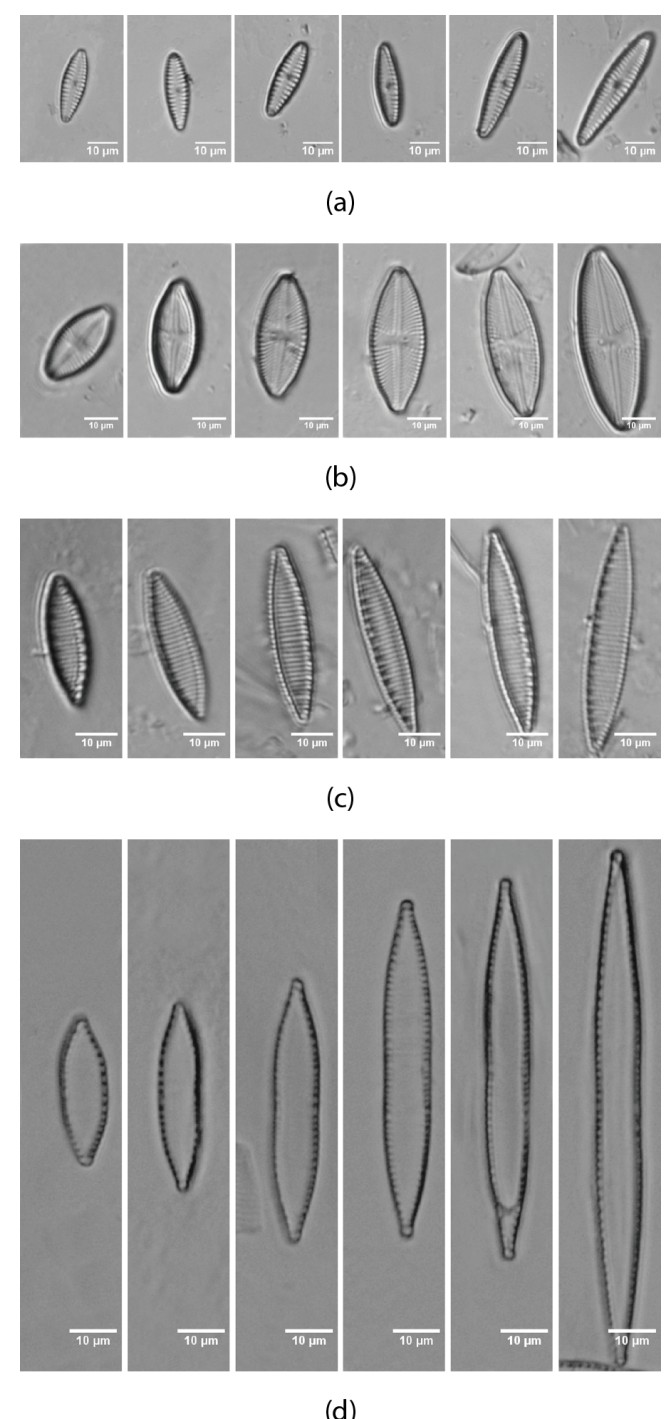

(a)

(b)

(c)

(d)

**Figure 2** **Life cycle of the diatoms present in the dataset (Table 1) from AQUALITAS project.** (A) *G. minutum*, (B) *L. goeppertiana*, (C) *N. amphibia*, (D) *N. capitellata*.

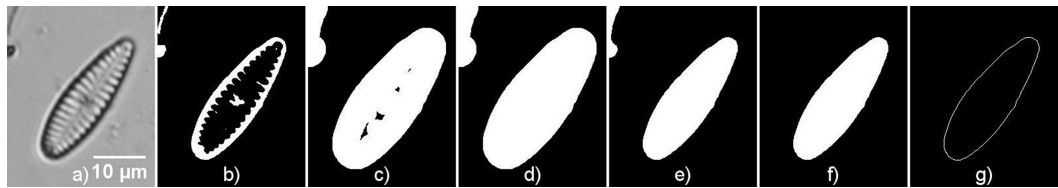

**Figure 3** **Contour extraction steps.** (A) Original image (*G. minutum*), (B) Binarized image, (C) Dilated image, (D) Image after filling holes, (E) Eroded image, (F) Biggest region, (G) Contour of the diatom.

5.  Selection of the biggest region.
6.  Contour extraction.

Figure 3 shows an example of the segmentation and contour extraction.

## Feature extraction

Three different descriptors have been chosen for this study. Elliptical Fourier descriptors (EFD) were used to describe diatom contour. Phase congruency and Gabor descriptors were used to characterize the texture of the diatoms. All of them are combined to form a feature vector that is used for classification. This feature vector is too big for classification and clustering so the dimensionality of the space is reduced with Linear Discriminant Analysis (LDA).

### Elliptical Fourier descriptors

We obtain EFD of the contour using the method described by *Kuhl & Giardina (1982)*. An implementation of EFD is available in *BielStela (2017)*. Taking a contour image as the starting point, we calculate the Freeman chain code. Then being $a_i$ the *ith* element in the Freeman chain code we obtain:

$$\Delta x_i = sgn(6 - a_i)sgn(2 - a_i) \tag{1}$$

$$\Delta y_i = sgn(4 - a_i)sgn(a_i) \tag{2}$$

$$\Delta t_i = 1 + \left(\frac{\sqrt{2} - 1}{2}\right)\left(1 - (-1)^{a_i}\right) \tag{3}$$

$\Delta x_i$ and $\Delta y_i$ are the changes in the $x, y$ projections of the chain code. $\Delta t_i$ is the modulus of the segment between two points $i$ and $i+1$. View Fig. 4 for more information.

Then we calculate the Fourier coefficients of the $x$ and $y$ projections of the Freeman chain code. $a_n, b_n, c_n, d_n$ represent the $n$ harmonic Fourier coefficients and $T$ the perimeter.

$$a_n = \frac{T}{2n^2\pi^2} \sum_{p=1}^{K} \frac{\Delta x_p}{\Delta t_p}\left[cos\frac{2n\pi t_p}{T} - cos\frac{2n\pi t_{p-1}}{T}\right] \tag{4}$$

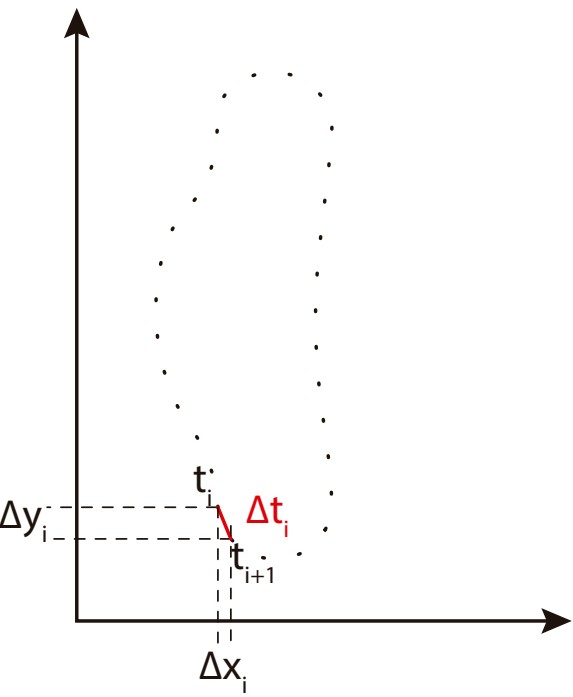

**Figure 4** Freeman chain code projections used to calculate Fourier descriptors (adapted from *Tort, 2003*).

$$b_n = \frac{T}{2n^2\pi^2} \sum_{p=1}^{K} \frac{\Delta x_p}{\Delta t_p} \left[ \sin\frac{2n\pi t_p}{T} - \sin\frac{2n\pi t_{p-1}}{T} \right] \qquad (5)$$

$$c_n = \frac{T}{2n^2\pi^2} \sum_{p=1}^{K} \frac{\Delta y_p}{\Delta t_p} \left[ \cos\frac{2n\pi t_p}{T} - \cos\frac{2n\pi t_{p-1}}{T} \right] \qquad (6)$$

$$d_n = \frac{T}{2n^2\pi^2} \sum_{p=1}^{K} \frac{\Delta y_p}{\Delta t_p} \left[ \sin\frac{2n\pi t_p}{T} - \sin\frac{2n\pi t_{p-1}}{T} \right]. \qquad (7)$$

Finally, it was empirically found that the first 30 coefficients provide an accurate approximation to the contour.

The amplitude of the $n$th harmonic can be calculated as:

$$amp_n = \frac{1}{2}\sqrt{a_n^2 + b_n^2 + c_n^2 + d_n^2} \qquad (8)$$

Figure 5 shows an example of the reconstruction of a contour using EFD when a different number of coefficients are used.
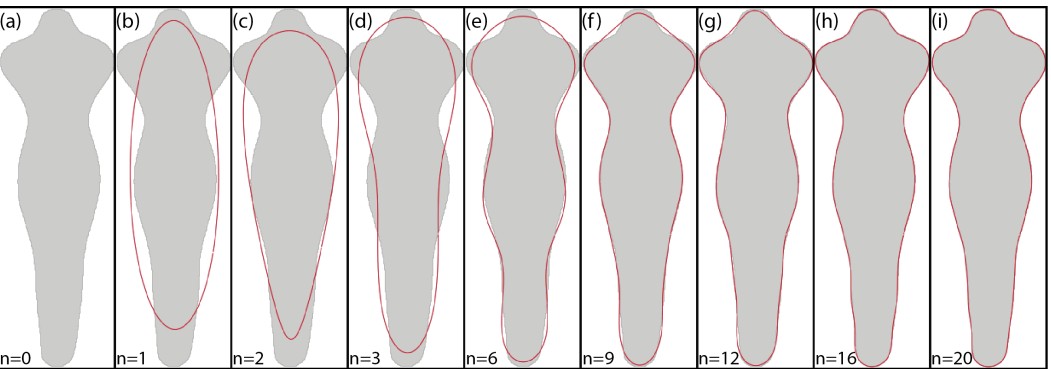

**Figure 5** **Example of EFD contour reconstruction with different order descriptors (A)–(I), where *n* represents the number of harmonics used.** Generated using the code available in *BielStela (2017)*.

### Phase congruency descriptors

The method of phase congruency ($PC$) is based on the concept that all Fourier components are in phase in the areas where signal changes occur. In the case of the images, these zones correspond to the edges, corners and textures of the objects. Therefore, the method seeks to obtain the maximum phase components in the Fourier domain. The main advantage of this method has to do with the fact that it is very robust to changes in lighting and contrast. This is due to the fact that the method works with the phases of the Fourier components but not with their amplitude.

Phase congruency was previously used as a preprocessing stage to contour extraction. *Sosik & Olson (2007)* found that a simple threshold-based edge detection applied to phase congruency provides excellent results to an ample set of phytoplankton images. *Verikas et al. (2012)* applied preprocessing through phase congruency-based methods for the enhancement of image edges. Phase congruency, as well as the gradient operator, is sensitive to noise. *Li et al. (2006)* performed a comparison of phase congruency with Canny edge detector and observed that the former is able to extract more detailed information than the later. They applied a contour length filter to remove the noise. In this paper, PC was not applied for contour improvement but for the extraction of discriminant descriptors.

Phase congruency (PC) based descriptors are calculated as in *Verikas et al. (2012)* by computing mean and standard deviation from phase congruency maximum ($M$) and minimum ($m$) momentum images (described in *Kovesi, 2003* and available in *Kovesi, 2000*). Those images combine the phase congruency information of each orientation. At the end, we have four phase congruency descriptors.

The phase congruency is obtained according to the following expression:

$$PC = \frac{\sum_\theta \sum_n w_\theta(x) \lfloor A_{n\theta}(x) \Delta\Phi_{n\theta}(x) - T_\theta \rfloor}{\sum_\theta \sum_n A_{n\theta}(x) + \epsilon} \tag{9}$$

$$\Delta\Phi_{n\theta}(x) = \cos(\phi_{n\theta}(x) - \overline{\phi}_{n\theta}(x)) - |\sin(\phi_{n\theta}(x) - \overline{\phi}_\theta(x))| \tag{10}$$
where $\theta$ indicates the orientation, $A_{n\theta}(x)$ and $\phi_{n\theta}(x)$ the amplitude and phase angle respectively used with the frequency component $n$, orientation $\theta$ and location $x.\overline{\phi}_\theta$ is the amplitude of the average phase angle in the orientation $\theta$, $w_\theta$ a frequency parameter in the orientation $\theta$ and $\epsilon$ a constant to prevent division by zero. $T_\theta$ is the noise estimated in the orientation $\theta$ that must be suppressed.

Maximum and minimum momentum images are calculated as in Eqs. (11) and (12) as a function of the $PC$.

$$M(x) = \frac{1}{2}[c + a + \sqrt{b^2 + (a-c)^2}] \tag{11}$$

$$m(x) = \frac{1}{2}[c + a - \sqrt{b^2 + (a-c)^2}] \tag{12}$$

where $a$, $b$, $c$ are:

$$a = \sum_\theta [PC_\theta(x)\cos(\theta)]^2 \tag{13}$$

$$b = 2\sum_\theta [PC_\theta(x)\cos(\theta)][PC_\theta(x)\sin(\theta)] \tag{14}$$

$$c = \sum_\theta [PC_\theta(x)\sin(\theta)]^2 \tag{15}$$

where $PC_\theta(x)$ is the phase congruency value at orientation $\theta$:

$$PC_\theta(x) = \frac{\sum_n w_\theta(x)\lfloor A_{n\theta}(x)\Delta\Phi_{n\theta}(x) - T_\theta\rfloor}{\sum_\theta \sum_n A_{n\theta}(x) + \epsilon} \tag{16}$$

Figure 6 shows an example of a diatom and its corresponding $m$ and $M$ images.

### Gabor filters

Gabor based descriptors are calculated by the same method as in *Bueno et al. (2017)* and originally described in *Fischer et al. (2007)*. The implementation can be found in *Cristobal, Fischer & Redondo (2016)*. First we calculate the log-Gabor filters as Gaussians shifted from the origin at different scales, $s$, and orientations, $t$, and they are applied to the input images. The formulation of the log-Gabor filters is:

$$G_{(s,t)}(\rho,\theta) = exp\left(-\frac{1}{2}\left(\frac{\rho - \rho_s}{\sigma_\rho}\right)^2\right)exp\left(-\frac{1}{2}\left(\frac{\theta - \theta_{(s,t)}}{\sigma_\theta}\right)^2\right) \tag{17}$$

where $(\rho,\theta)$ are the log-polar coordinates, the number of scales is $S=4$ and the number of orientations is $O=6$. Thus, $s \in \{1,\ldots,S\}$ and $t \in \{1,\ldots,O\}$ indexes the scale and the orientation of the filter, respectively. And $(\rho_s, \theta_{(s,t)})$ are the coordinates of the center of the filter; $(\sigma_\rho, \sigma_\theta)$ are the angular and radial bandwidths in $\rho$ and $\theta$ (see *Fischer et al., 2007*) for more details.

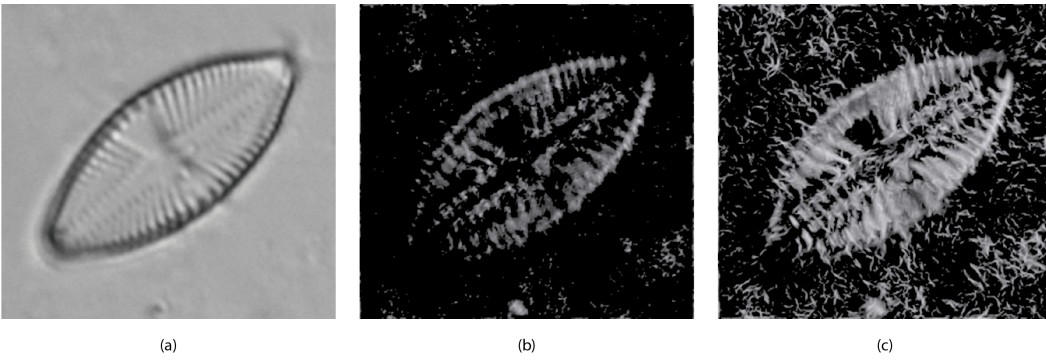

**Figure 6** (A) Original image (*L. goeppertiana*) (B) Minimum momentum of phase congruency image (m). (C) Maximum momentum of phase congruency image (M).

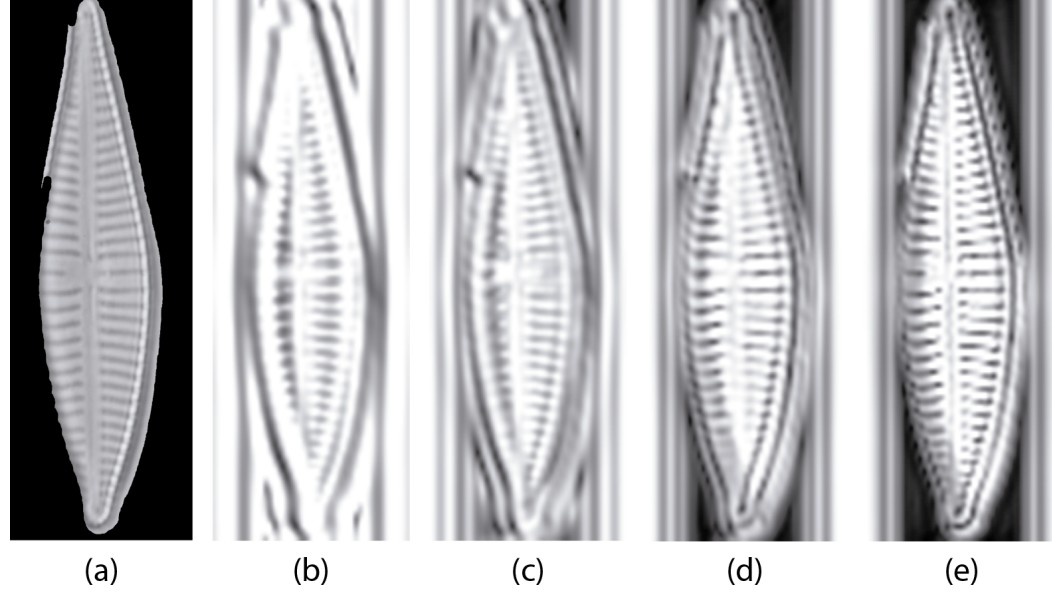

(a)  (b)  (c)  (d)  (e)

**Figure 7** Example of diatom image after the log Gabor filters ($L = 4, O = 6$). (A) is the original masked image (*G. jadwigiae*), (B)–(E) are the four bands applied to the image. For visualization purposes images a log function was applied for scaling (B)–(E).

Then first and second order statistics are acquired for every sub-band. Gabor feature reduction was obtained using correlation for removing redundant information. With this reduction procedure we finally obtain a 177 Gabor feature vector from the original 1460 one. Figure 7 shows an example of log-Gabor filtering (four bands. G1, G2, G3 and G4) applied to a diatom example.

During the feature selection process other alternative descriptors were considered. We did some experiments with SURF features (*Bay, Tuytelaars & Van Gool, 2006*) but the results did not improve the final performance. Figure 8 presents the results of applying the

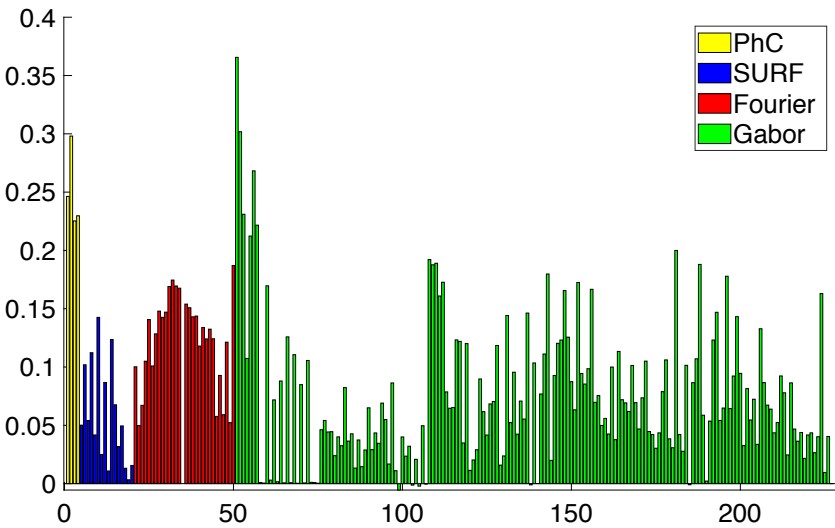

**Figure 8** **Importance of the different features obtained from Relieff algorithm** (*Robnik-Šikonja & Kononenko, 2003*). Abcissa axis represents number of descriptor. Ordinate axis represents Relieff value. Relieff values have been calculated using Matlab `relieff` function.

Relieff algorithm to the full set of features selected, including SURF. Due to the low scores of SURF such features were discarded.

## Dimensionality Reduction

As a result of applying all the feature extraction algorithms, a 211 dimensions feature vector was generated. It has been shown in Fig. 8 that there are more discriminative features than others. With that in mind, we can reduce the complexity of the feature vector and concentrate the discriminative power of all the features in a lower dimensionality space. In order to reduce the complexity of the vector for the subsequent classification, we need to perform a dimensionality reduction. Linear Discriminant Analysis (LDA) or Fischer Discriminant Analysis is a supervised method for dimensionality reduction described in *Fisher (1936)* although it also has been used as a classifier. Originally it was described for a 2-class problem and it was later generalized as a multi-class LDA by *Rao (1948)*. The main objective of LDA is to project the feature space into a new smaller subspace that maximizes the separation between classes. LDA reduces the dimensionality of the original number of features to $(N-1)$ features, where $N$ represents the number of classes.

## Classification

A classifier can be understood as a function that takes the extracted descriptors as input and by using different algorithms produces an output that represents the probability that a given characteristic belongs to a certain class (*Bueno et al., 2017*). More details on different classification algorithms can be found in *Alpaydin (2010)*.

Classifiers can be divided into two different classes: supervised and non-supervised. On the one hand the former group needs the data labeled, i.e., in the training stage it needs
to know the correct class for each of the samples. On the other hand, non-supervised classifiers infer the structure of the data from the unlabeled input.

Different supervised and non-supervised classification techniques have been evaluated for testing the extracted features. The implementation of these techniques is available in *Pedregosa et al. (2011)*.

### Supervised

An extensive range of supervised classifiers like nearest-neighbor, Supported Vector Machines (SVM), Random Forest and Bagging Trees, the quadratic Bayes normal classifier and the Fisher classifier were compared. The best results were obtained with KNN and SVM. Both algorithms need previous training. This training was carried out by selecting a small subset from the image dataset as training data and the rest as test data. Thus, a 10-fold cross-validation (10fcv) scheme was followed, where 10 image samples are used as the validation set and the remaining data as the training set. This was repeated $C_{10}^n$ times, where $n$ is the total number of images in each dataset, and $C_{10}^n$ is the binomial coefficient. This is done to divide the original sample on a validation set of 10 samples and a training set. Finally, the arithmetic mean of the results of each iteration was performed to provide a single and final result.

- **KNN**: refers to K-Nearest Neighbors (*Friedman, Bentley & Finkel, 1976*). This classification algorithm assigns a class to each sample, choosing between the class of the $K$ nearest neighbors, giving a confidence value of the assigned class. The distance between different elements can be computed as a simple Euclidean distance. KNN is one of the most common and straightforward classification methods. The procedure is highly dependent on the value of $K$ which is usually determined empirically.
- **SVM**: refers to Support Vector Machine. It was first proposed by *Vapnik (1999)*. This method determines the hyperplane that best divides the data into the different classes. The algorithm increases the dimensionality of the features space, so a non-linear classification problem can be solved linearly. The distance between the hyperplane and the training data is called the functional marging, which is used as a confidence interval of classification results (*Wang et al., 2017*).

### Non supervised clustering

Three different clustering algorithms were selected for non-supervised classification: K-means, Hierarchical Agglomerative Clustering and BIRCH. Although the methods are labeled as unsupervised, they can actually be considered as semi-unsupervised because the number of clusters was identified with the number of classes.

- **K-means**: K-means algorithm (*Lloyd, 1982*) separates all the data into $K$ clusters where the distance between the data and the cluster centroid (mean of all the data in the cluster) is minimized. The centroids are initialized at random points and they are updated after a feature is assigned to a cluster. As a result of the algorithm, the data space is partitioned into Voronoi cells, i.e., a diagram that divides the space in a given number of regions. A related method to K-means is the so-called K-medoids (*Park & Jun, 2009*). Unlike the

K-means, K-medoids chooses datapoints (medoids) and uses squared Euclidian distance to define distance between data points.

- **Hierarchical Agglomerative Clustering** (*Manning, Raghavan & Schutze, 2008*): This algorithm starts with a cluster for each observation. These clusters are successively merged together, minimizing a distance function between the clusters. The process ends when the predefined number of clusters has been reached.
- **BIRCH**: refers to Balanced Iterative Reduced Clustering using Hierarchies (*Zhang, Ramakrishnan & Livny, 1996*). This algorithm is divided into four different phases. In phase 1 a Clustering Feature Tree (CF tree) is built with all the data. The second phase scans the CF tree to remove outliers and group crowded subclusters into larger ones to get a new smaller CF tree. This step is optional. Phase 3 is a global clustering procedure applied to the reduced CF tree. The last step is optional and it refines the clustering results. It obtains new clusters using the centroids of phase 3 as seeds for the clustering algorithm.

### Clustering validation metrics

In order to validate clustering results five different metrics were considered (*Vinh, Epps & Bailey, 2010*; *Kassambara, 2017*).

- **Adjusted RAND index**: This index measures the similarity between the clustering labels assignment and the given ground truth. Its value varies in the range $[-1,1]$, where 1 is perfect score.
- **Silhouette**: This metric evaluates the similarity between an element and others members of the same cluster compared to members of other clusters. This metric can indicate how compact a cluster is and how it is separated from other clusters. It can take on a value between $-1$ and 1. Measures close to 1 mean well defined clusters.
- **Adjusted Mutual Information** (**AMI**): This metric compares the agreement between clustering class assignments and the ground truth classes. It can take a value between $-1$ and 1 and values close to 1 indicate significant agreement.
- **Homogeneity**: It measures if a cluster consists only of member of the same class or not.
- **Completeness**: It measures if all the members of a class are assigned to the same cluster.

## RESULTS

The results section is divided into a few experiments to validate the proposed set of features as a proper method to describe the diatoms life cycle for automatic identification as well as to compare the results obtained with the proposed descriptors with other methods in the literature. The assessment of the descriptors will be carried out using different supervised and non-supervised classifiers. For supervised classifiers, a k-fold cross-validation procedure has been used in order to reduce possible biased results. These experiments evaluate both the discriminant power of the features between different diatom genera and between different species of the same genera.

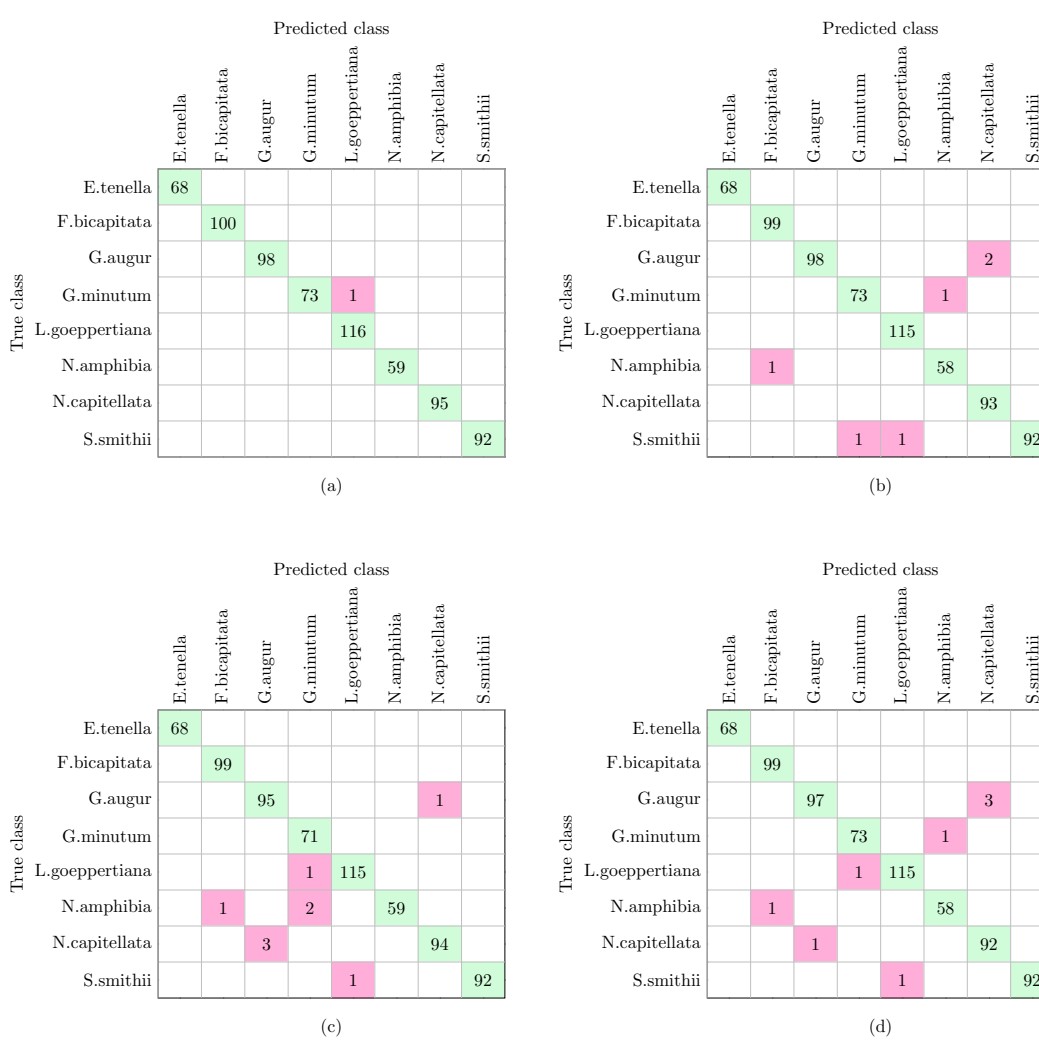

**Figure 9 Confusion matrices of supervised and non supervised classifiers.** (A) SVM and KNN. (B) K-means. (C) Hierarchical Agglomerative clustering. (D) BIRCH.

## Experiment 1

The complete workflow proposed in this work (Fig. 1) was tested using the dataset shown in Table 1. The dataset analyzed in this experiment and described in Table 1 is the only one of the three datasets that include more morphological variability in the species due to the presence of different stages in the diatom life cycle. This characteristic together with the fact that more taxa are analyzed leaded to a reduction in the classification accuracy in comparison with Experiments 2 and 3 (see below). With this dataset SVM and KNN classifiers were able to correctly identify all the specimens except only one that was missclassified as is shown in Fig. 9A. This figure represents a confusion matrix, i.e., the elements located along the main diagonal represent correct classifications while those who fall outside represent errors. On the one hand, from the confusion matrix it can be concluded that supervised methods achieve a 99.9% accuracy in identification.
On the other hand, non supervised methods, namely K-means, Hierarchical Agglomerative and BIRCH clustering provide 99.1%, 98.9% and 98.7% accuracy respectively, with nine being the largest number of errors. Analyzing the confusion matrices (Figs. 9B–9D), the most repeated error was classifying *G. augur* as *N. capitellata*.

In a previous study focused on diatom curvature analysis, (*Wishkerman & Hamilton, 2017*) showed that LDA performed better that PCA in separating taxa. However, they recommend using both PCA and LDA because the analysis is data dependent. In all the experiments described here, LDA outperformed PCA by providing more compact and disjoint clusters. Figures 10A and 10B show how the clusters are distributed in the feature space for both PCA and LDA applied to the extracted features respectively. It is important to notice that the percentage of accumulated variance by the first two dimensions of the vector is higher for LDA resulting in better separated clusters. In this figure, the first two dimensions (out of seven) of the resulting feature vector are presented with the Voronoi regions of the calculated clusters. Despite having only two components in the graph, as they concentrate most part of the variance of the dataset, a very good separability between the clusters is observed. Clustering performance can be evaluated through the previously defined metrics. Figure 11 shows the values of the metrics for the three clustering algorithms. The values provided by the metrics are in line with the accuracy results, with K-means providing the best result and BIRCH the worst. RAND and AMI can be also interpreted as an accuracy measure. In this case, both measures give higher values than 97% for the three described clustering algorithms. The values of homogeneity and completeness indicate that the clusters only contain members of their class and that all the members of a class have been assigned to the same group. None of the clustering methods provide perfect accuracy due to the presence of some identification errors. Finally, the Silhouette coefficient indicates the degree of compactness and how close or far the clusters are. In our case, with a value above 0.5 provided by all methods, it can be concluded that separation and compactness is not optimal but it is enough to discriminate between the different classes.

## Experiment 2

In the previous experiment all the diatoms in the dataset were from a different genera, therefore there were a lot of dissimilarities between them (Fig. 2). When studying diatoms from the same genera but different species the similarities between specimens increase. In this experiment, the dataset is formed by 479 images of six different species of *Sellaphora* (Table 2). In *Mann et al. (2004)* the authors used Principal Component Analysis (PCA) with their descriptors (the first nine even Legendre polynomial coefficients). PCA is a dimensionality reduction algorithm that transforms the feature vectors into another space where the first components accumulate most of the variance. Figure 12 shows the difference between the result of applying the procedure described by *Mann et al. (2004)* and the method described in this work using LDA as the dimensionality reduction procedure for both features set. LDA was used instead of PCA (*Mann et al., 2004*) descriptors in order to make a fair comparison. Figure 12B shows a greater separation and grouping of the

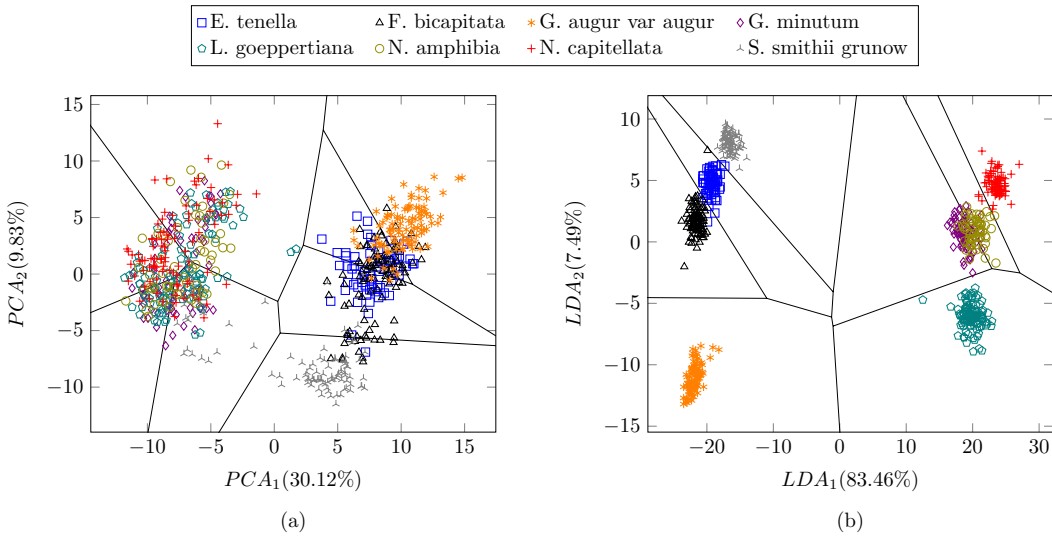

**Figure 10** **Representation of the clusters and first two components of the resulting feature vector after dimensionality reduction of the Table 1 dataset using (A) PCA and (B) LDA.** For visualization purposes centroids of the clusters were calculated with k-medoids. Note how the chosen features and classification methods allow in (B) to segregate into disjoint clusters even though the classes include life cycle related morphological variability.

samples than in Fig. 12A. With this dataset a classification rate of 99% is obtained. In *Mann et al. (2004)* classification rates were not provided.

## Experiment 3

In this experiment we tested the same dataset presented in (*Blanco, Borrego-Ramos & Olenici, 2017*) with the proposed method, i.e., 244 different valves corresponding to five diatom species of the same genera. In this case, the use of shape and texture improves the classification results up to 100% accuracy with all the classifiers described in this study. Figure 13 shows a perfect cluster separation. In *Blanco, Borrego-Ramos & Olenici (2017)* the authors concluded that only morphometric measurements based on taxonomic keys such as the length/width ratio or the striae density are not sufficient for diatom classification when classes are similar. They obtain correct classification rates between 40% and 70% when classifying six different species of the *Gomphonema* genera (see Table 3) with different classifiers and clustering algorithms.

## DISCUSSION

Automatic identification is a problem that has been the subject of different studies during the last years (*Hicks et al., 2006*; *Kloster, Kauer & Beszteri, 2014*; *Bueno et al., 2017*; *Pedraza et al., 2017*). This interest has recently increased ssince diatoms are a very good bioindicator of water quality, e.g., (*European Committee for Standardization, 2004*). This directive establishes that it is necessary to identify at least 400 valves in each sample prior to calculate
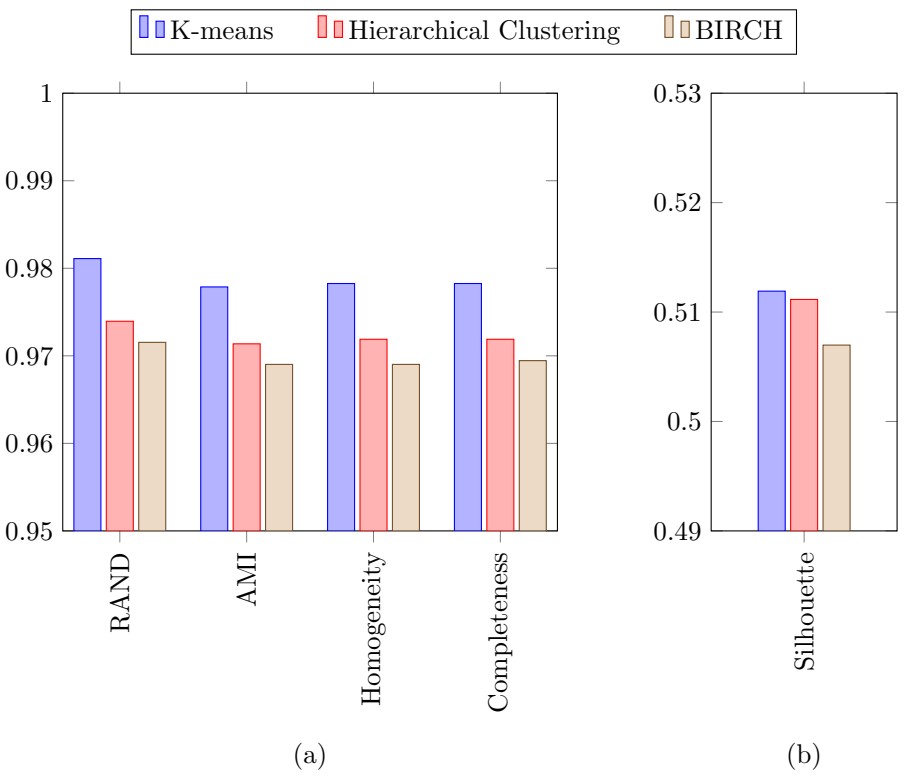

**Figure 11** **(A) RAND, Adjusted mutual information, Homogeneity and Completeness metrics and (B) Silhouette metric corresponding to the three selected clustering algorithms.**

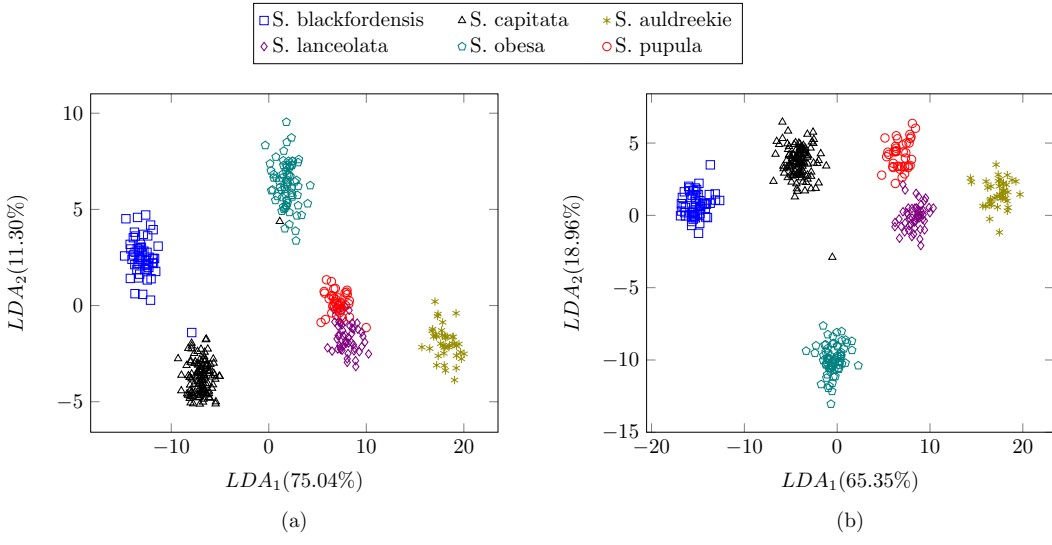

**Figure 12** **Comparison between (*Mann et al., 2004*) (A) process and the described method (B).**

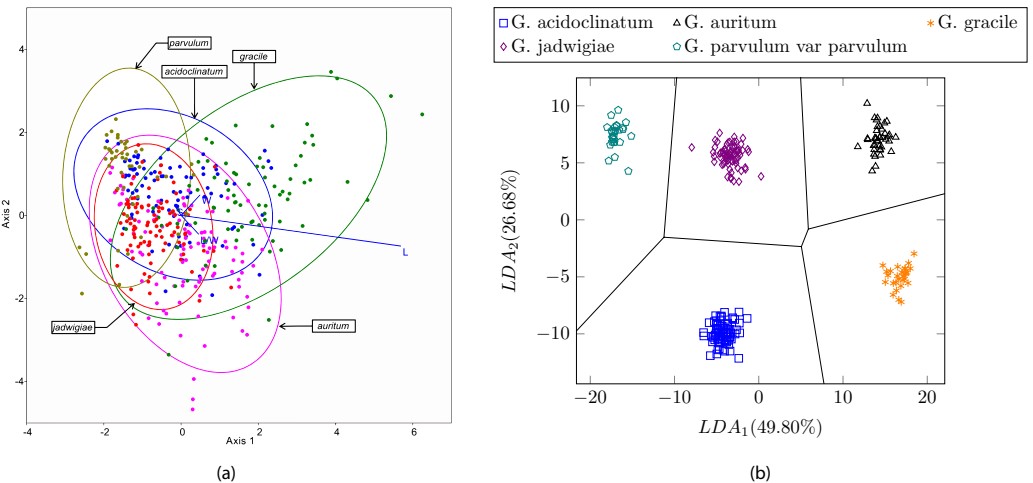

**Figure 13** **(A) Canonical variates analysis (CVA) biplot obtained in** *Blanco, Borrego-Ramos & Olenici (2017)*. **Dots represents individuals and lines predictors. (B) Cluster representation for the first two components of the resulting feature vector after dimensionality reduction corresponding to the dataset presented in** Table 3. For visualization purposes centroids of the clusters were calculated with K-medoids.

the water quality index. By automating this process the productivity of the experts will increase.

In this work, an automatic method of diatom identification is presented. According to previous studies (*Blanco, Borrego-Ramos & Olenici, 2017*), it was shown that morphometric measurements are not sufficient for automatic diatom identification. Therefore we selected a group of descriptors that combine morphometric and texture information to improve the degree of discriminability. With this set of descriptors, we increased the results from 70.2% accuracy obtained by Blanco et al. to a 100% accuracy for five different classes and higher than 97% with other testing datasets. This result proves that the combination of chosen features and classification methods achieve high accuracy levels and outperform methods based on morphometric measurements.

There exists very few datasets publicity available that explicitly include diatom's life cycle. One of the reasons may have been due to the difficulty of capturing a sufficient number of specimens of each species in each of the stages of its life cycle. Dataset 1 includes few species extracted from the DIADIST and AQUALITAS projects that explicitly include the life cycle variation. Dataset 2 corresponds to a previous study of the Sellaphora pupula species complex by *Mann et al. (2004)* that implicitly includes the life cycle variation. The reason for including such dataset was to perform a comparison study of the discriminant power of the feature descriptors used by *Mann et al. (2004)* and ours. Dataset 3 was recently analyzed by *Blanco, Borrego-Ramos & Olenici (2017)* and the reasons for its analysis were the same as the previous case. Dataset 1 explicitly includes different stages of the life cycle of the diatoms, which increases the degree of intraclass variability. This characteristic together with the fact that more taxa are analyzed led to a reduction in the classification accuracy in comparison with Experiments 2 and 3. The diatoms in dataset 1 are of the

same genus but of different species narrowing the degree of intraclass variability. One of the main criteria when selecting the discriminant features was the possibility of obtaining high classification rates in both scenarios. According to the results obtained, rates higher than 97% are achieved and therefore such requirement is fullfilled.

With dataset 1 we obtained classification accuracy of 99.9% with supervised classifiers and an average of 98.9% with non-supervised classifiers for eight classes. In this case, there is a small gap between both supervised and unsupervised learning. When this happens, non-supervised learning is usually preferred since there is no need for training and it reduces the manual work done by the experts (e.g., data labeling for training). Despite the good results obtained, the database used is not very big and it is possible that the difference between supervised and unsupervised classifiers increase, making supervised preferable over non supervised in the case of large databases. It is left as future work the elaboration of a bigger dataset focused in diatoms life cycle and to test the proposed method with this new dataset. It will also be interesting to analyze if the percentage of data correctly classified is maintained or it decreases with the addition of new classes. With datasets 2 and 3, classification rates of 99.76% and 100% respectively were obtained. In the case of dataset 1 that includes more morphological variability due to the presence of different stages in the diatom life cycle, a reduction in the classification accuracy was observed although this fact needs to be corroborated in the future by considering a large number of species.

More recently other authors use different techniques (*Pedraza et al., 2017*) approaching the automatic diatom identification problem using deep learning techniques. A 99% overall accuracy was obtained by the authors using a diatom dataset with 80 classes. It is not easy to compare the results of the aforementioned experiments with those presented here due to the different nature of the datasets. While dataset 1 has only eight classes it is focused on having samples representing the different stages of the life cycle of every class. The use of deep learning (DL) techniques is out of the scope of the current paper. The main problem would be to have available large datasets for training what in the case of diatoms constitutes a major difficulty. Previous work by the authors (*Pedraza et al., 2017*) shown that a minimum number of 300 samples/class (valve) is needed so that DL techniques improve the results that can be obtained with handcrafted techniques for diatom classification (*Bueno et al., 2017*).

As *Mann (2018)* has recently pointed out, a future line of research of interest that has received little attention would be related to the study and quantification of the deformation of the girdle throughout the different stages of the life cycle and its relation with the changes in shape.

Another interesting topic when using automatic identification is how reliable the system is when it is compared with an expert. *Culverhouse et al. (2003)* highlights the difficulties faced by human experts when the identification task involves similar specimens with little variation between them. The study concludes that trained experts can obtain between 67% and 83% accuracy while experts that perform identification tasks as a routine obtain identification accuracy in the range of 84% to 95%. With automatic methods, such as those described in this work, the precision of the experts can be greatly improved.

The life cycle is not the only source of shape and ornamentation changes for diatoms. There are other environmental factors that can induce changes in diatom morphology producing abnormalities called teratologies. *Falasco et al. (2009b)* made a review of diatom teratologies and their origins. In this review, the authors conclude that diatoms are very sensitive to environmental conditions like changes in the pH of the water, presence of heavy metals or other toxic compounds, etc. Therefore the use of such teratological forms appears to be very important for environmental studies. Testing the proposed classification procedure with these abnormal forms is out of the scope of this study, and it is left as future work.

## CONCLUSIONS

The main purpose of this study was to establish a set of robust image descriptors able to describe diatoms including life cycle stages. To this end, we combine shape and texture information of the diatom valves to achieve automatic identification of diatoms when the diatom life cycle was considered. According to the obtained results, the selected features and classification methods provide very good performance for the task of diatom categorization. Both supervised and non-supervised classifiers obtained good accuracy results up to 99.9% and 98.9% respectively. It was also shown that LDA is preferable (vs. PCA) as a dimensionality reduction technique for multivariate classification.

Finally, it is necessary to highlight the importance of capturing the morphological variability derived from the life cycle in the training set for improving the identification accuracy which is a consequence of the diversity that one sees in natural habitats.

## ACKNOWLEDGEMENTS

The authors thank Saúl Blanco and María Borrego-Ramos from Institute of the Environment in León for providing images to perform the experiments.

### Funding

This work was supported by the Spanish Government under the Aqualitas-retos project (Ref. CTM2014-51907-C2-2-R-MINECO). The funders had no role in study design, data collection and analysis, decision to publish, or preparation of the manuscript.

### Grant Disclosures

The following grant information was disclosed by the authors:
Spanish Government under the Aqualitas-retos project: Ref. CTM2014-51907-C2-2-R-MINECO.

### Competing Interests

The authors declare there are no competing interests.

## Author Contributions

- Carlos Sánchez performed the experiments, analyzed the data, contributed reagents/materials/analysis tools, prepared figures and/or tables, authored or reviewed drafts of the paper, approved the final draft.
- Gabriel Cristóbal conceived and designed the experiments, analyzed the data, contributed reagents/materials/analysis tools, prepared figures and/or tables, authored or reviewed drafts of the paper, approved the final draft.
- Gloria Bueno performed the experiments, analyzed the data, contributed reagents/-materials/analysis tools, authored or reviewed drafts of the paper, approved the final draft.

## Data Availability

The raw data is available at Figshare: Diatom life cycle images dataset

Blanco, Saul (2018): Diatom life cycle images dataset. figshare. Dataset. DOI: 10.6084/m9.figshare.7077725.v2.

Gomphonema life cycle images dataset

Blanco, Saul (2018): Gomphonema life cycle images dataset. figshare. Dataset. DOI: 10.6084/m9.figshare.7146518.v1.

## Supplemental Information

Supplemental information for this article can be found online at http://dx.doi.org/10.7717/peerj.6770#supplemental-information.

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
