# Peer review of "Diatom identification including life cycle stages through morphological and texture descriptors"

_PeerJ, doi:10.7717/peerj.6770_

## Round 0.1 · original submission · Major Revisions

· Academic Editor

Major Revisions

I have heard back from three qualified reviewers, and they all agree your work is of interest and importance to the scientific community. However, all three have offered constructive comments to help you improve your manuscript. While none of the comments alone suggest the need for a major revision, there are many helpful comments you will wish to consider, and thus my decision is major revisions. Having read the work myself, I also agree that the English needs some editing here and there; please see the specific comments of reviewers. Also, please check capitalization of species names in Figures 9, 10, and 12. I look forward to seeing a revised version.

·

Basic reporting

The manuscript is written in a clear language, it is easy to read and to follow. I am a biologist working with diatoms and I had to skip the maths in the Methods section as I am not qualified to judge those parts. I am glad to see that the image sets used are publicly available. The same is not true about analysis code for segmentation and feature extraction or any downstream analysis – as a diatomist, I can only recommend to the authors to consider the possibility of publishing their analysis code (especially relating to segmentation and feature extraction) since the features extracted can be highly interesting for many diatomists, but hardly any of them will be able to apply them only having the mathematical descriptions.

I have one main problem with clarity of the manuscript: I am finding that the first sentence of the conclusions (l. 324-326), or the line of argumentation in the discussion quite clearly reflect the contents of the manuscript, but the Introduction and the Abstract (and even the title) do not do so quite as clearly. The way I see it is that the main novelty of this paper is a) the set of feature descriptors that it uses, and b) that it uses (at least one of the three datasets analyzed?) sets of images explicitly capturing the range of life cycle / size related morphological variability in the target taxa. As the Discussion and the Conclusions clearly state, the features extracted are used in algorithmic classification experiments. To the contrary, the title, abstract and introduction focus on life cycle modelling, a related, but different aspect. The manuscript does not provide any explicit statements about life cycle models inferred by any of the models, only about classification accuracy and discriminating power of features. One can of course argue that a classification algorithm trained on a set of images depicting life cycle related variation must implicitly model the life cycle, but to me it would make the situation clearer to call these analyses classification (in the statistical meaning) or identification (in the biological meaning) experiments. For this reason, I would suggest the authors to consider re-formulating the title, introduction and abstract, de-emphasizing the aspect of life cycle modeling and making it clear up front that what is being done is image classification.

I am finding the third paragraph of the introduction (l. 43-65) somewhat convoluted. The text rightly points out difficulties with modelling diatom life cycles (non-linearity and multidimensionality), but then the manuscript does not in any explicit manner address these problems. The text then gives a short literature review of many things related to what one could call the large topic of digitally capturing/extracting morphological information from diatom images, but in a to me, not very logical ordering. The authors could consider separating the issues of feature extraction for diatom morphometric characterization from capturing life cycle related variation and explicitly modelling the latter.

I am missing citations pointing to precise description of applied methods (for instance, the whole section on classification methodology from line 173 to 220 only includes two references, both relating to support vector machines, but none to any of the other applied techniques).

The statement that currently more than 100’000 diatom species are described (l. 24) appears rather high to me, please cite a reference on which this number is based. Also the second paragraph of the Introduction (l. 31-42) comes out without a single citation – probably no problem for a diatomist but it could be useful to cite some sources, to other potential readers not so familiar with the organisms.

It did not become quite clear to me what the differences between the analyses applied to data set 1 vs data set 2 and 3 were. Reading the Results and the Discussion, it seems that a) only data set 1 is explicitly meant to capture life cycle related variation – is this correct? If yes, it would be useful to explicitly state this, perhaps in the Methods section; b) in experiments 2 and 3, the final (supervised and unsupervised classification) steps of the procedure shown in Figure 1 were not applied – is this correct? If yes it would be good to state this explicitly in the Methods.

Experimental design

Experimental design is sound and I do not have any major criticisms related to it, only some remarks for consideration.

One methodological novelty of the paper is the use of LDA for dimensionality reduction (instead of the commonly used PCA). If this is really a novelty (as it seems since no previous examples are cited), it would be interesting to see a comparison of how this difference affects classification performance. One further remark related to this aspect is that the paper does not specify how many dimensions were kept after the LDA for downstream analyses.

The manuscript applies a workflow to three low diversity diatom image data sets in three separate experiments (8, 6 and 5 taxa, respectively). The discussion then states (l. 295-299) that the size of the data sets is rather small and testing the methods on more “realistic sized” data sets remains to future work. It appears to me that it would have been a simple extension of the study to pool the three image sets for a fourth experiment. This is still far from the diversity one sees in most natural habitats, but would nevertheless go one step further into that direction. I would encourage the authors to consider adding such an analysis if feasible.

Validity of the findings

To me the main (potential) findings of the paper are that

a) the proposed features do a good job in capturing morphological information separating different diatom species, this is convincingly supported;

b) LDA is preferable (vs. PCA) as a dimensionality reduction technique for multivariate classification – this point is not addressed in a systematic manner; if that is an intention of the authors, it would be informative to compare at least two analyses which only differ in this aspect, such a comparison is currently not found in the paper;

c) since the aspect of life cycle is given a lot of sentences, especially in the first half of the paper, one would expect to also find a conclusion that relates to this topic, but I do not really see such a finding. One might say that it is important to capture life cycle related morphological variability in the training set in automatic identification attempts, but I currently do not see anything in this paper that would make this point clearer or better supported than it already was. Perhaps this is only a question of clearer formulation in the discussion or a more explicit comparison between data set 1 vs. 2 & 3 (the former apprently capturing life cycle more completely than the latter two)?

Additional comments

The final paragraph of the introduction could be removed since it only outlines the standard manuscript structure common to the majority of PeerJ articles.

The use of parentheses in citations is unusual and inconsistent, for instance double parentheses on l. 28 or in the legend of Fig. 8; parentheses between author and year instead of before author name on l. 45, 54, 55, 59-63 (although this variant can of course be used if the author name is part of the sentence like on l. 50 or 57).

Specific epithets are capitalized inFig. 9, 10 and 12 but not in 13 – I would recommend to not capitalize them.

Figure 8 in its present form is completely incomprehensible to me; although a reference is cited, it would help the reader if the idea behind this plot was shortly explained.

I would interpret Figure 11 to say that the difference between the three clustering methods is marginal but this might be caused by the scaling of the y-axis – would it make sense to separate out Silhoutte (which lies in a different range from the other metrics) and plot all metrics with narrower y-axis limits?

Related to the same comparison, how was the final number of clusters chosen in the clustering exercises? If it is simply chosen as the known number of classes in each data set, then this cannot be called a completely unsupervised classification approach since the number of clusters present in an unlabeled data set is in general not known a priori.

The sentence on l. 200-201 seems incomplete or is otherwise incomprehensible to me.

Fig 12 compares newly generated with previously published data from identical sets of images. We see strongly improved separation of a priori known classes, but there are multiple differences between the old and new analyses which makes it difficult to pin down whether the improved results are caused by the novel set of features used vs. the different dimensionality reduction technique (PCA vs. LDA). That the latter improves separation is not surprising; it would be more interesting to see the effect of the novel features alone, like in Fig. 13, in which case it could make more sense to present LDA results on both the old and the new feature set side by side.

Reviewer 2 ·

Basic reporting

Basic reporting

The paper is easy to read, but a major text revision is required for correction of mistakes. I have found word-choice, misspelling and punctuation errors. In addition, some phrases should be rewritten. Several examples were the language should be improved include lines:
41,150 Contractions are inappropriate for scientific papers.
45-46: Blanco et al. (2017) --> (Blanco et al., 2017). There are some more cases like this.
48-50: phrase is very long and can be made more understandable by adding some comma or rewritten it.
91: pues Images: text written in Spanish and the first letter of “images” written in capital letter.
103: dilatation --> dilation
110: Phase Congruency descriptors and Gabor descriptors --> Phase congruency and Gabor descriptors.
129: the amplitude of them --> their amplitude
132: combine all the information of phase congruency --> combine the phase congruency information:
135-138 where θ indicates the orientation, … and φ are the… --> where θ indicates the orientation, … and φ the… The use of is, are can be avoided here.
170: On the other hand, but you must also use “on the one hand” to describe different ideas. Do the same with similar expressions
236: On the other side --> on the other hand, but “on the one hand “is also missing.

The first letter of the name of a figure or table must be written in capital letter, for example:
81 "As Fig. 1 shows"
93 “Images from Table 1”

Experimental design

The experiment design seems OK; however, I do not think the Otsu method provides a good segmentation in every image. I would like to know if I am wrong or what was done when images were wrongly segmented.
Line 177: “training was carried out by selecting a small subset” --> Please, specify how many images or which percentage was used for the training.

Validity of the findings

The results seem good. however, the Otsu algorithm does not always produce a good segmentation of diatom images, which is important to ensure adequate contours. Likewise, phase congruence does not always provide adequate results if images are noisy or diatom are not clearly defined. Therefore, it would be appropriate to know if these cases were presented and how they were avoided or resolved.

264-265: The percentages of the variance along the axes are higher in LDA than in the case of PCA --> This is only true on the abscissa axis.

Additional comments

I feel that the authors should revise their manuscript, even though the results seem to be good. I therefore suggest to rewrite some parts of the paper, improving the quality of the text

Reviewer 3 ·

Basic reporting

The paper proposes a method for classification of diatoms an its life cycle modeling. Features based on elliptic Fourier descriptors, phase congruency and Gabor filters are used in two supervised and three unsupervised classification methods. Three unsupervised clustering methods are compared with five validation metrics. Three datasets are considered.
Clear English language is used throughout the paper. Introduction describes the main objectives and related work is sufficiently covered. However, some equations have unclear notation or variables are undefined. See my general comments below.

Experimental design

Methods and experiments are described with sufficient detail, however the relevance to diatom life cycle modeling is not well stated. See my general comments below.

Validity of the findings

Experimental results demonstrate that on the given dataset the chosen features and classification methods achieve high accuracy levels and outperform methods based on morphometric measurements. Unfortunately, comparison with a deep learning technique was not carried out due to the different nature of the datasets. See my general comments below.

Additional comments

1) The algorithm for mask extraction is described in lines 101-106. It seems to me that it is designed to work only if one single diatom is present in the image. If the system is presented as an automatic method of diatom identification (line 284) then I think it is necessary that it can also work with images that do not contain any diatoms and also that contain more than one.

2) It is not clear how many features were used in experiments after dimensionality reduction. In line 241 a number seven is mentioned, yet it is not clear if this is the dimension of the feature space. I suspect that the number of selected features is important and therefore the authors should include accuracy results w.r.t. to the number of features or at least some discussion on this topic.

3) In Experiment 2, the same features are used and the only difference is whether LDA or PCA is used for dimensionality reduction? If this is true, it should be clearly stated.

4) The experiment 1 considers diatoms from a different genera and both experiments 2 and 3 consider diatoms from the same genera (lines 255-257). However the achieved accuracy is higher in the exp. 2 and 3 than in the exp. 1, which is not intuitive. It is indicted later at lines 291-294 that the dataset in exp. 1 includes diatom life cycles, which make the diatom classes more variable and therefore decreases classification accuracy. I suggest to clearly state in the section “Experiment 1” the presence of diatom life cycles. Life cycles are mentioned in the title, but experiments do not discuss this phenomenon. It would then be clear why the exp.1 accuracy is lower then in exp.2 and 3. I also suggest to improve in this manner the section “Discussion”.

5) It is unfortunate that the comparison on the dataset (80 classes) used in Pedraza et al. (2017) could not be done. It would further strengthen this paper.

6) Notation in some equations is not clear.
PC is defined in eq. (9) and then used in eqs. (13)-(15). Theta (orientation) is used as a parameter but in (9) theta is integrated out. Is PC in (9) a different PC in (13)-(15)?
In the definition of Gabor features in eq. (16), the function B(u,v) is not defined.

Minor:
- line 91: “pues” not an English word.

---

## Round 0.2 · Minor Revisions

· Academic Editor

Minor Revisions

I have heard from the same three reviewers as in the first round of reviews. All are very positive, and reviewers 1 and 2 have some small comments that need addressing. I imagine these revisions will be easy to do, and anticipate a quick acceptance of the next version of this work provided you respond to their comments.

·

Basic reporting

Substantially, the authors added pointers to publicly available code relating to their analysis, I welcome that.
Also citations in the Methods section were added.
I commented previously on the statement that “currently more than 100’000” diatom species are described. The authors now added two sentences detailing different estimates for possible number of diatom species; I do not see these as necessary, my point was simply to say that the number of currently described diatom species (which I think should be in the 10thousands) was confused with the estimated total number (to which the figures given by the authors refer) in the original version. This has now been fixed.
The main line of argumentation in the Introduction has improved; only one comment: a sentence was now added saying “the main source of errors come 59 from the misclassification of algae through their life cycle” – if this is so, that would be a quite substantial insight for the field, but what is this statement based on? No citation is given.

Experimental design

Here I previously asked the authors to clarify how novel their use of LDA instead of for dimensionality reduction is in a supervised classification framework. This was not addressed, neither the quantification of how/to what extent this difference affects classification performance. What is given instead is a graphical comparison between a PCA and an LDA ordination which of course shows a better visual separation in the second case (but which is not a great novelty or surprise). I would not like to push this issue further, only remark that if the authors see this as a substantial methodological innovation, it would be good to put it into a more solid (quantitative and literature) context.
My other comment (reflecting on the discussion point related to the lack of availability of higher diversity data sets) was also not addressed on l.107-120, as the authors state, but that is perfectly fine, it was only a suggestion.

Validity of the findings

My previous comments under this point have more or less been addressed

Additional comments

The minor points I raised here previously were all addressed

Reviewer 2 ·

Basic reporting

There is a big improvement with respect on the previous version of the paper. There are only few small typographical or grammatical errors to be corrected.Some examples are:

Line 46: where it not be reduced ->where it cannot be reduced.

Line 155 gradient operator will be sensitive -> gradient operator is sensitive
Line 156: edge Canny edge -> Canny edge
Line 158 PC was not applied for contour improvement but it was used for -> PC was not applied for contour improvement but for

Line 169 where PC is the phase congruency -> and Pc is the phase congruency.

Experimental design

No comment.

Validity of the findings

No comment.

Additional comments

I think the paper is good for publication. It is very well written and easy to understand. However, there are still some grammatical mistakes to be corrected.

Reviewer 3 ·

Basic reporting

no comment

Experimental design

no comment

Validity of the findings

no comment

Additional comments

The authors have improved the manuscript and I am satisfied with their responses to my comments.

---

## Round 0.3 · accepted · Accept

· Academic Editor

Accept

I have read through your rebuttal to comments and the revised manuscript, and it is now acceptable for publication. I look forward to seeing your final published work!

One note - please change "presence of diatom life cycles" to "presence of different stages in the diatom life cycle" (both Introduction and M&M).

#